# Effect of Virtual Reality Therapy on Quality of Life and Self-Sufficiency in Post-Stroke Patients

**DOI:** 10.3390/medicina59091669

**Published:** 2023-09-15

**Authors:** Marcela Dąbrowská, Dalibor Pastucha, Miroslav Janura, Hana Tomášková, Lucie Honzíková, Šárka Baníková, Michal Filip, Iva Fiedorová

**Affiliations:** 1Department of Epidemiology and Public Health Protection, Faculty of Medicine, University of Ostrava, 703 00 Ostrava, Czech Republic; hana.tomaskova@osu.cz; 2Department of Rehabilitation and Sports Medicine, Faculty of Medicine, University of Ostrava, 703 00 Ostrava, Czech Republic; miroslav.janura@osu.cz (M.J.); lucie.honzikova@osu.cz (L.H.); sarka.banikova@fno.cz (Š.B.); michal.filip@osu.cz (M.F.); iva.fiedorova@fno.cz (I.F.); 3Department of Rehabilitation and Sports Medicine, University Hospital of Ostrava, 708 52 Ostrava, Czech Republic; 4Faculty of Physical Culture, Palacký University Olomouc, 779 00 Olomouc, Czech Republic

**Keywords:** virtual reality, quality of life, activities of daily living, stroke

## Abstract

*Background and Objectives*: The consequences of stroke have a significant impact on self-sufficiency and health-related quality of life (HRQoL). Virtual reality (VR)-based rehabilitation has the potential to impact these modalities, but information on timing, volume, and intensity is not yet available. The aim of this randomized controlled trial (1:1) was to evaluate the impact of conventional rehabilitation combined with VR on self-care and domains of HRQoL in patients ≤6 months post-stroke. *Materials and Methods*: The intervention group completed a total of 270 min of conventional VR + rehabilitation sessions. The control group underwent conventional rehabilitation only. Primary assessments with the WHO disability assessment schedule 2.0 (WHODAS 2) questionnaire were conducted before rehabilitation (T0), after completion of the intervention (T1), and at the 4-week follow-up (T2); secondary outcomes included self-sufficiency and balance assessments. *Results*: Fifty patients completed the study (mean age 61.2 ± 9.0 years, time since stroke 114.3 ± 39.4 days). There were no statistically significant differences between the groups in WHODAS 2, self-sufficiency, and balance scores (*p* > 0.05). *Conclusions*: In the experimental group, there was a statistically significant difference in WHODAS 2, assessment of self-sufficiency, and balance scores before and after therapy (*p* < 0.05). VR appears to be a suitable tool to supplement and modify rehabilitation in patients after stroke.

## 1. Introduction

Stroke is considered a worldwide epidemic [1], the consequences of which are associated with high mortality, severe disability [2], functional impairment, and cognitive deficits [3,4]. Stroke can also have a significant impact on the quality of life (HRQoL) and independence of patients [5]. The recent literature shows that more than 40% of survivors six months after a stroke have residual functional impairment, 30% have limited self-sufficiency, only 10% survive completely without sequelae [6,7], and 25% remain in institutional care [8]. The increasing number of patients with severe residual deficits has a long-term negative impact on health and social systems.

Given the lack of evidence on the effect of VR on HRQoL and self-sufficiency in post-stroke patients, we have focused our study on evaluating the effect of VR in combination with conventional therapy. The primary objective of our randomised trial was to evaluate the effect of conventional rehabilitation combined with VR on improving self-sufficiency and HRQoL in patients after an ischaemic stroke.

Researchers suggest that there is still a lack of relevant information in clinical practice on the optimization and appropriate timing of regular individual interventions in the daily context of stroke care [9]. Coordinated rehabilitation is an integral part of comprehensive therapy [10]. Rehabilitation of post-stroke patients is a dynamic process [11] that involves evaluation, identification, and quantification of patient needs. Then, it is followed by the definition of a realistic and achievable goal and a targeted intervention [12]. This cyclical, progressive, goal-oriented process enables the person with a disability to achieve optimal levels of physical, cognitive, emotional, communicative, social, occupational, and functional activity [11,13]. With a reduced level of self-sufficiency, the patient loses functional independence in activities of daily living and employment and becomes dependent on the help of others. Up to 57% of post-stroke patients require help with personal activities of daily living (ADL) required for biological needs such as personal hygiene, bathing, dressing, self-feeding, use of the toilet, and control of urination and defecation [14]. Training in ADL increases patient independence and therefore the possibility of returning to the home environment, which in turn improves patient HRQoL and reduces financial cost of care [15,16]. Effective rehabilitation programmes augmented with virtual reality can also contribute to improved independence and HRQoL [1].

Virtual reality (VR) is a technology that can mimic the real environment through a computer and allows users to interact with this environment [17]. VR is increasingly becoming an integral part of medicine and is one of the new neurorehabilitation approaches in the treatment of motor and cognitive dysfunctions in post-stroke patients [1]. VR features, such as immersion, imagination, and interaction, can induce enjoyment and motivation [18] in the patient to perform more meaningful activities. Virtual reality training may then present some advantages over conventional training, as it allows for individualised repetitive movements while stimulating motor and cognitive processes [19]. Motor learning through VR facilitates the effective implementation of learnt skills in the real world [1,16]. It allows the therapeutic process to extend throughout the spectrum of rehabilitation, from acute inpatient settings to post-acute facilities to follow-up and self-therapy in the home environment after discharge. The results of meta-analyses and systematic reviews that have evaluated the effect of VR therapy (on motor function of the upper extremities, cognitive function, HRQoL, and ADL) in post-stroke patients provide heterogeneous results [1,5,17,20]. Khan et al. (2023) reported that virtual reality is an effective tool in the rehabilitation of post-stroke patients, particularly in motor function, but no statistically significant differences were found compared to standard therapy [17]. Laver et al. (2017) published that VR therapy alone is not more beneficial than conventional therapeutic approaches to improve upper limb function [1]. However, its combination with conventional therapy may be useful in improving upper extremity function and activities of daily living [21]. Other studies confirm that VR training is more motivating, enjoyable, and engaging than conventional therapeutic exercises [22,23,24] as conventional therapies relate to routine care. Therefore, patients may lose motivation and adherence to treatment as therapeutic movements become tedious and monotonous over time [25].

## 2. Materials and Methods

### 2.1. Study Design

A randomised controlled monocentric study was conducted in a population of patients after their first ischaemic stroke in the arteria cerebri media basin who were admitted to a rehabilitation sanatorium. The study was approved by the Ethics Committee of the University of Ostrava Faculty of Medicine. All patients signed an informed consent.

### 2.2. Participants

The study included patients after their first ischaemic stroke in the arteria cerebri media basin (time since the event <6 months) who were referred for rehabilitation for 4 to 5 weeks in a rehabilitation sanatorium in the Czech Republic, between November 2022 and March 2023. Patients were treated according to the European Stroke Organization (ESO) standard operating procedure based on the Grading of Recommendations, Assessment, Development, and Evaluations (GRADE) framework. All patients were treated pharmacologically prior to the start of rehabilitation, and during their stay in the rehabilitation facility, they underwent regular medical examinations at least once a week.

The inclusion criteria for the study were those aged 40–79 years, in stable condition, with ability to participate, mini-mental state examination score <25 points, intact vision, preserved grip function of the thumb and index finger of the affected limb, and functional mobility according to Functional Ambulatory Category (FAC) 3–5.

Study exclusion criteria included: age >40 and <79 years, decompensated state, cardiovascular instability, severe fatal and severe cognitive impairment, low functional mobility according to FAC 0–2, dementia with mini-mental state examination >24 points, severe visual impairment, and personal history of epilepsy. The selected patients had no other neurological disease that could affect the measurement results. All selected patients signed an informed consent to be included in the study. The study was approved by the Ethics Committee.

### 2.3. Procedures

All patients underwent an initial examination (T0) by a trained occupational therapist on the first day after admission to the rehabilitation sanatorium. The mini-mental state examination (MMSE), the Barthel index (BI), the extended Barthel index (EBI), the Berg balance scale (BBS), and the WHO disability assessment schedule 2.0 (WHODAS 2.0) tests were performed. Mobility was assessed according to the FAC scale (0–5) [26]. All tests were performed in a quiet environment with no distractions. Demographic and clinical data were obtained from medical records and patient interviews (age, sex, education, occupation, time since the stroke, type of stroke, and stroke therapy). After baseline measurements, they were randomised into intervention and control groups. Further measurements with MMSE, BI, EBI, BBS, and WHODAS 2.0 (T1) (assessment) were performed after the completion of the intervention, and a follow-up assessment (T3) was performed 4 weeks after the completion of rehabilitation.

MMSE is the most widely used 10-item test to detect cognitive deficits, evaluating the patient’s orientation to time and space, short-term memory, numeracy, attention, speech, and constructive practical skills [27,28]. MMSE does not assess judgment, logical thinking, or vocabulary, but it does a good job of differentiating moderate dementia from normal ageing. The maximum score is 30 points, and at ≤24 points, the dementia band begins [27,28].

BI is a tool used to assess functional capacity, the ability to perform basic activities of daily living after a stroke [29,30]. BI is a valid measure of activities of daily living and is considered a simple and rapid tool for measuring ADL [31]. It assesses 10 activities of daily living such as food intake, bathing, personal hygiene, dressing, bladder continence, rectal continence, toilet use, transfers, locomotion, and stairs walking, which can be scored at 0–15 points. The total score range is 0–100 points [28,32,33]. A final total score of 0–20 points reflects very severe dysfunction, a score of 25–45 points reflects severe dysfunction, a score of 50–70 points reflects moderate functional impairment, a score of 75–95 points reflects mild functional impairment, and a score of 100 points reflects full self-care of ADL [31].

EBI helps to assess the need for assistance in performing ADL where cognitive ability is a prerequisite [34,35]. The test assesses the following elements: comprehension, communication, social interaction, everyday problem solving, memory, learning and orientation, vision, and neglect syndrome. Each of the six items is scored on a 0–5 scale (0—worst performance and 5—best performance), with a maximum of 15 points per item. The maximum score Is 90 points. The extended Barthel test categorises patients into three categories: severe cognitive deficit (0–15), moderate cognitive limitation (20–65), and no or mild cognitive deficit (70–90) [34,35].

BBS assesses the patient’s static and dynamic balance [36]. BBS consists of 14 elements that contain basic components to analyse the ability to maintain balance during standing, standing on one leg, sitting, transfers, and rotations that are necessary for activities of daily living [37,38]. Individual items are scored on a scale of 0 to 4 (0—no performance, 4—highest score). A total score of 41–56 points predicts a low risk of falling, a score of 21–40 points predicts a moderate risk of falling, and scores of 20 or less are indicative of a high risk of falling [37]. 

WHODAS 2.0 is an international standardised generic test that evaluates disability as defined by the International Classification of Functional Abilities [38,39]. WHODAS 2.0 is designed specifically for patients after stroke and has excellent psychometric properties. Validation studies have demonstrated good internal consistency of the test, high validity, good sensitivity to changes over time, and sensitivity to capture functional changes in treatment groups [38,39]. WHODAS 2.0 includes six core quality of life domains. It allows a subjective assessment of the patient’s condition, the determination of their disability, and the subsequent identification of their needs, necessary for prioritisation, determination of therapy, and selection of adequate resources [37,38]. Domain 1 assesses understanding and communication, domain 2 assesses mobility, domain 3 deals with self-care, domains 4 and 5 include relationships with people and assess life activities, and domain 6 focuses on participation in society. Domain scores and total scores can be calculated using the SPSS algorithm (Statistical Package for the Social Sciences) algorithm [40,41]. The item-response theory is applied in the scoring, where individual items may have different weights. Disability is expressed as a percentage from 0% to 100% [40,41].

### 2.4. Instruments Used

Oculus Quest 2 (Meta Quest 2) has 6 GB of RAM combined with the ultra-fast Qualcomm Snapdragon XR2 platform. It consists of a projection helmet and two controllers. The helmet houses two screens for the eyes with the ability to adjust focus and the addition of an eyeglass attachment. A fast switchable LCD display with low backlight persistence, 1832 × 1920 pixels per eye resolution, will allow visual fidelity of approximately 21 PPD (pixels per degree) without blurring or doubling artefacts. Adjustable straps are used to secure the helmet to the head. Both the right- and left-touch controllers have multiple control buttons, a joystick, and an adjustable fixation strap that attaches to the wrist to prevent the controller from potentially falling out of the hand. With built-in Bluetooth and an app on a mobile phone or tablet, the therapist can monitor what the patient sees and how effectively they perform each task.

### 2.5. Intervention

In addition to conventional therapy, the patients in the intervention group received training using the Oculus Quest 2 VR device. In total, there were a minimum of 10 and a maximum of 15 therapy sessions of 20 min per day, 3 times per week for 4 to 5 weeks. The total volume of active VR therapy averaged 270 min for the entire intervention. When determining the 20-min unit of time, we based this on the amount of time the patient spends with a physical or occupational therapist during the exercise unit. The standard therapeutic unit lasts 30 min, but this does not deduce the time the patient takes to talk to the therapist and assess their current condition, remove clothing and shoes, and assume a suitable position for exercise. The first three therapy sessions, which were aimed at familiarising the patient with the functions of the system controls using the First Steps for Quest 2 programme, lasted a maximum of 10 min. The patients were seated for the first two sessions and completed the third and subsequent sessions standing. From the fourth therapy, VITALIS Pro VR 0.4.1 software was used as part of the intervention. All patients completed the exercises in a total of 4 programmes: free painting, 2D tracing, 3D painting, and puzzles. In the “free painting” programme, they were asked to draw specific shapes; in 2D and 3D, they traced the shapes shown. The “puzzle” programme had three levels of difficulty and a hint option. The patients performed each task in three types of environments, namely forest, space, and sea. Each environment also conveyed different sound sensations to the patients. When performing tasks in the forest environment, patients heard birds singing; in the sea environment, it was the sound of waves; and in the space environment, it was computer music. Throughout the exercise, the study coordinator was present in the room and monitored the patient for symptoms indicating sudden instability, seizure, shoulder, arm, or hand pain. If the patient did not feel well at any time during the exercise, the coordinator instructed them to stop the task. Patients enrolled in the study received exercise in VR due to organizational reasons of the rehabilitation facility in the afternoon from 3 h after the completion of all treatments.

As a part of conventional therapy, patients were indicated for individual neurophysiological-based physiotherapy 2 times a week for 30 min (total 240–280 min per stay). The other part consisted of occupational therapy aimed at targeted training of self-sufficiency, gross and fine motor skills, monomanual and bimanual activities, and improvement of movement coordination. Occupational therapy was performed twice a week for 30 min (240–280 min in total per stay). Furthermore, patients received twice-weekly bath with iodine–bromine + wrap, oxygen therapy, whirlpool bath, pool exercise, dry hot compresses, classical massage, mechanotherapy, CO_2_ gas injections, four chamber bath, and electrical stimulation.

### 2.6. Statistical Methods

Basic descriptive statistics and statistical tests according to the type of data were used for statistical evaluation. The normality of the data was tested using the Shapiro–Wilk test, followed by a two-sample t-test and a non-parametric Mann–Whitney test, paired Wilcoxon test, and Spearman correlation coefficient. For qualitative data, the chi-square test was used; if the conditions for its use were not met, Fisher’s exact test was used. Statistical tests were evaluated at the level of significance of 5%. Stata version 17 was used for processing.

## 3. Results

We recruited 75 patients for the study and a flow chart of the study participants is presented in accordance with the CONSORT requirements (Figure 1). Of the 75 patients approached, 50 met the entry and exit criteria (*n* = 50, 26 M, 24 F, mean age 61.2 ± 9.0), completed follow-up tests, and completed the WHODAS 2 questionnaire 4 weeks after the end of therapy. Twenty-five respondents were excluded from the study due to premature termination of stay in a rehabilitation facility due to illness, positivity for COVID-19, or failure to complete the quality of life questionnaire 4 weeks after the end of therapy.

The experimental group consisted of 25 patients (mean age 59.4 ± 8.9 years, 12 women, 13 men); the control group consisted of 25 patients (mean age 63 ± 8.8 years, 12 women, 13 men). There were no statistically significant differences in baseline sociodemographic and clinical parameters between the two groups (Table 1). A negative correlation between age and the Barthel index output was found in the entire cohort, but for men and women separately, this relationship was no longer statistically significant.

Table 2 shows the comparison of the results of the input and output tests of MMSE, WHODAS 2, BI, EBI, and BBS between the two groups. The input values were part of the inclusion criteria for the study. In the experimental group, there was no statistically significant change in any of the tests applied after treatment compared to the control group.

Table 3 shows the results of the quality of life assessment using the WHODAS 2 questionnaire.

Figure 2 shows the results of the quality of life assessment using the WHODAS 2 questionnaire between the experimental and control groups as shown in Table 2.

There were no significant differences between the experimental and control groups in the individual domains of HRQoL and self-sufficiency tests. Statistically significant differences were found in all domains for each treatment group. All patients in the experimental and control groups consistently reported that their self-sufficiency improved after rehabilitation. These positive effects were found after therapy and persisted four weeks after therapy. The greatest limitations in mobility were reported by the patients in items that evaluated standing endurance for longer periods of time, for example, 30 min, and walking for longer distances, for example, 1 km. The patients in the experimental group reported better abilities in managing daily activities such as showering, dental hygiene, and dressing. For women, problems with putting on underwear, specifically fastening bras, persisted. After rehabilitation, patients continued to have limitations in ADL when handling small objects, such as unlocking doors. In ADL, both sexes consistently reported deficits in self-care, specifically manipulating cutting and cutting. Both genders also consistently reported concerns about staying home alone for several days.

Cognitive impairment is a common sequela after stroke that can persist and affect ADL performance, especially in the early stages after stroke. In the domain of evaluation of cognitive function, patients perceived persistent mild to moderate deficits in attention and focus on activity for 10 min and also in remembering important things and learning new things. In the domain of assessing interpersonal relationships, patients described persistent mild to moderate deficits in dealing with strangers and making new friends. Moderate to severe difficulties in the domain of sexual activity persisted in patients even four weeks after the completion of rehabilitation. Participation and involvement in social activities were also rated as severe by most patients in both experimental and control groups. Women were more likely to report moderate and severe emotional difficulties due to their health condition.

## 4. Discussion

New sophisticated virtual reality technologies implemented in the therapy of post-stroke patients have gained importance in recent years and are promising forms of technology that have been shown to increase patient satisfaction with post-stroke rehabilitation [17,41,42]. The intent of this study was to test whether adding low-cost Oculus Quest 2 virtual reality technology to conventional therapy in a rehabilitation facility setting would effectively improve the self-sufficiency and HRQoL of post-stroke patients.

In the studied patient group, statistically significant changes occurred in all domains of the quality of life and self-sufficiency questionnaire after VR-assisted therapy, and this change persisted four weeks after the end of therapy. Although all patients subjectively rated VR therapy as beneficial, motivating, and enjoyable, and the experimental group experienced improvements in their self-sufficiency, quality of life, and stability, statistical significance was not demonstrated when the test results were compared with those in the control group. Patients perceived and described VR exercise as effective, as immersion increased their motivation to complete meaningful activities, while completion of tasks allowed them to increase the intensity of purposeful repetitive movements. Self-sufficiency in activities that require coordinated hand and arm actions, such as dressing, shoeing, and showering, improved for all patients. After therapy, patients persisted with deficits in fine manipulative activities such as grasping and manipulating keys, coins, and buttons, which require good finger–thumb coordination, which may have been influenced by the programme and controllers not allowing repeated finger involvement in fine motor functional activities during the activity. Virtual reality programmes are designed to simulate objects and events in the real world [1] and therefore can induce intrinsic motivation, especially if they contain game elements [42,43]. When patients are excited about their activities and experiences, they are immersed in the game and are often more motivated to complete them even when performing otherwise uninteresting activities [42,43,44]. Therefore, virtual reality and interactive video games can not only motivate patients but may have some advantages over conventional rehabilitation approaches, as they provide opportunities to practise everyday activities and learn new skills that cannot be practised in a hospital setting [1]. In conventional rehabilitation programmes, in a relatively stimuli-free environment, patients often perceive repetitive physical activities as boring, unattractive, unexciting, and less motivating [45]. Reduced motivation may result in fewer functional repetitions of movements, which does not produce a targeted effect in rehabilitation. Zheng et al. (2015) monitored quality of life in stroke patients using the SF-36 questionnaire. They concluded that upper limb function, ADL, and quality of life can be improved in patients with hemiplegia in the subacute period by using VR exercises [46,47]. The VR therapeutic unit was carried out for 30–60 min, six times a week for 4 weeks. In this study, patients completed three times more VR exercises than patients in our experimental group who were rehabilitated with VR three times a week for 20 min for 4–5 weeks.

The patient’s ability to adapt to his/her new situation is also related to age, gender, education, and social class. Mobility and activities of daily living are the basis of the quality of life in patients after stroke. Performance in ADL becomes automated over the course of life, as activities are performed daily from a very early age, while the repertoire of ADL performance is shaped over the course of life, which also affects different cognitive and physical demands [47,48]. Due to automation, cognitive demands on personal ADL can be very low, while physical demands may change with age due to age-related declines in physical abilities, age-related changes, and increased comorbidities [47,48]. Wurzinger et al. (2021) reported that older age significantly affects ADL dependence 3 to 12 months after stroke. Some studies have found that age does not have a negative effect on the quality of life in patients after stroke [48], while others have found that age has a strong effect on the quality of life in patients after stroke [49]. Feigin et al. (2014) in their large study that included data from 119 countries pointed out that there is an increasing incidence of stroke in the younger population, with the number of cases in the age group of 20–64 years increasing from 25% in 1990 to 31% in 2010 [50]. Patients aged 40 to 79 years were included in our study; the youngest patient who received VR training was 40 years old and the oldest was 75 years old. A negative correlation was found between age and Barthel index output for the entire cohort, but for each sex separately, this relationship was no longer statistically significant. In studies, individuals with a BI score <95 are considered recovered and self-sufficient [47]. However, as shown in our study, post-stroke patients who achieved a BI score <95 continued to have residual deficits in ADL and IADL and impaired quality of life. Lin et al. (2023) used BI to assess ADL in their study and found that both experimental and control groups showed significant improvement over time, but there was no significant difference between the groups. BI is considered a valid and reliable tool for the assessment of stroke patients and is often used in the Czech Republic to assess activities of daily living in the acute care setting [47], but we recommend the selection of another more sensitive assessment tool for future studies.

The level of education, the level of comorbidity, as well as socioeconomic situation, occupational status, and potential return to work are also related to the quality of life. In our study, most patients were living in a relationship with a partner and all patients planned to return to their home environment. Hayes et al. (2003) reported in their study that up to 25% of patients after stroke remain in institutional care, and in their analyses, women were more dependent for ADL, less likely to go unassisted, and lived in nursing homes [7]. Hu et al. (2017) concluded that domain values of cognition level, mobility, and WHODAS 2.0 sum score were good predictors of institutionalization [50]. The authors of the studies also point out the influence of local social welfare policy, disparities in social support, and disability service system [50]. Lee et al. (2022) in their study reported that perceived social support is positively correlated with motivation to recover [19]. The higher the perceived social support, the stronger the stroke patient’s motivation to recover and the higher the likelihood of lower disability rates [19].

However, as new VR technologies are increasingly being implemented in patient rehabilitation programmes and studies report that the use of VR in stroke rehabilitation contributes to improved motor function, self-sufficiency, muscle strength, and balance, further studies are needed to ensure that VR programmes are appropriately selected, properly timed, and produce a targeted treatment effect [1,17,47].

The advantage of this study was the fact that the study had follow-up trials to monitor longer-term effects. Even after 4 weeks of therapy, the patients continued to experience a positive effect of therapy. The patients subjectively rated the therapy as motivating and effective. One limitation for our study is that some patients found VR tasks easy to perform after a certain period and would need incorporation of more challenging tasks into their therapy. Furthermore, the small sample size, which became even smaller due to the patients’ illness and the positivity of COVID-19, may also have affected the study results. The time since the stroke was not the same in all patients. The appropriate timing of physical activity is a topic of ongoing debate. Some patients prefer to remain temporarily at home after the stroke and postpone subsequent rehabilitation. This is a natural reaction of the patient who is forced to react to the change in his/her life situation, and his/her reaction depends on many factors, the degree of impairment of physical and mental functions, life experiences, personality, and attitude. Even spare capacity in subsequent rehabilitation facilities plays a role in early rehabilitation.

## 5. Conclusions

The application of VR did not lead to significantly different changes compared to conventional therapy. Patients experienced improvements in self-sufficiency and quality of life, but these improvements were not different from the changes seen with conventional therapy. However, more randomised controlled trials are needed to determine the effectiveness of VR training on the quality of life and self-sufficiency in post-stroke patients.

## Figures and Tables

**Figure 1 medicina-59-01669-f001:**
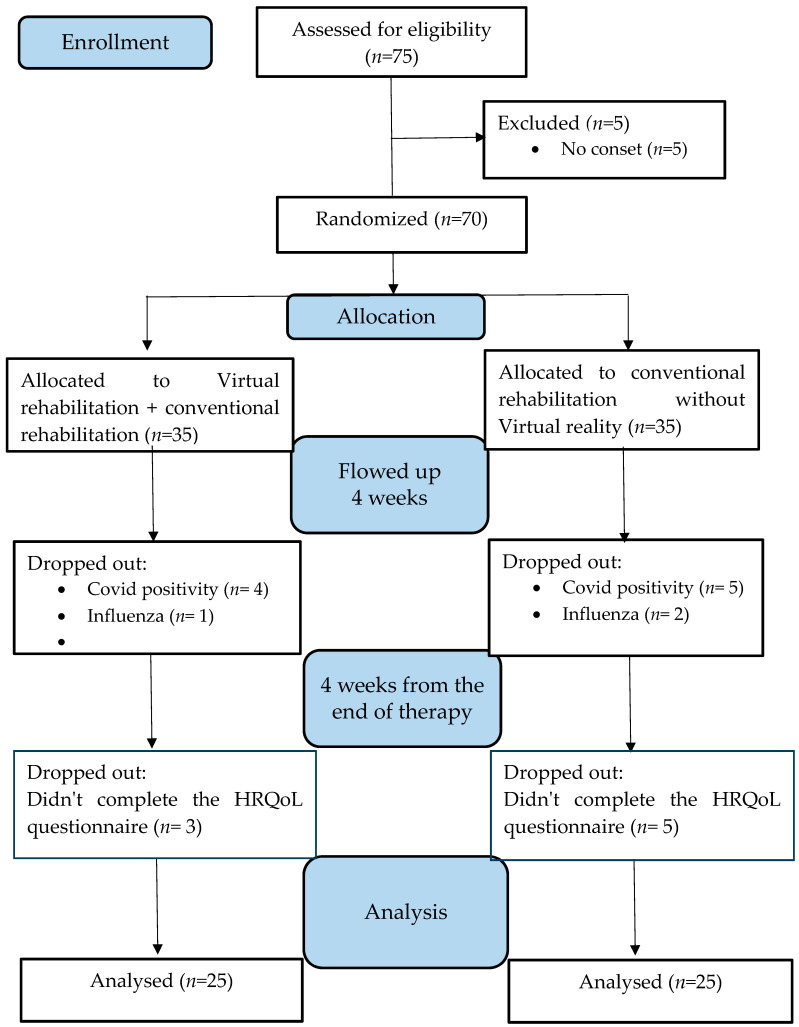
Flow chart of study participants according to CONSORT.

**Figure 2 medicina-59-01669-f002:**
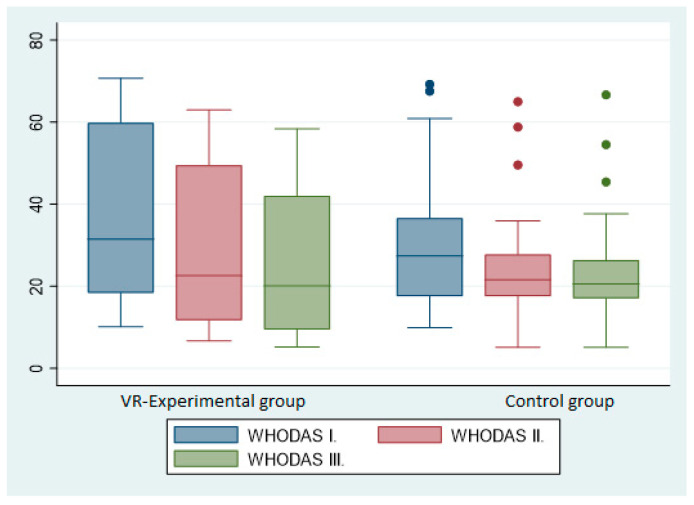
Explanatory notes: WHODAS 2.0—WHO Disability Assessment Schedule 2.0. The *x*-axis includes the experimental and control groups for each period T0, T1, T2. The *y*-axis includes the total percentage score of all domains of the WHODAS 2 questionnaire.

**Table 1 medicina-59-01669-t001:** Basic sociodemographic and clinical characteristics of the experimental and control groups and their comparison.

	General Characteristics of the Subjects		
Characteristics	Experimental Group (*n* = 25)	Control Group (*n* = 25)	*p*
Age (years), median (IQR)	59.36 (40–75)	62.96 (49–79)	0.158
Gender (Male/Female) (%)	13 (52)/12 (48)	13 (52)/12 (48)	1
Limb dominance side (right/left) *n* (%)	24 (96)/1 (4)	23 (92)/2 (8)	1
Side of the lesion (right/left) *n* (%)	17 (68)/8 (32)	14 (56)/11 (44)	0.561
Time since stroke (days), median (IQR)	126 (60.5–154)	128 (94.0–152.5)	0.762
Previous medical history, *n* (%)			
Hypertension	25 (100)	22 (88)	0.235
Diabetes	8 (32)	9 (36)	0.765
Hyperlipidaemia	20 (80)	15(60)	0.217
Ischaemic heart disease	4 (16)	2 (8)	0.667
Low back pain	18 (72)	14 (56)	0.239
Dysarthria	8 (32)	5 (20)	0.333
Obesity	13 (52)	11 (44)	0.571

Explanatory notes: IQR—Interquartile range, *p*–value.

**Table 2 medicina-59-01669-t002:** Comparison of MMSE, WHODAS 2, BI, EBI, and BBS median (IQR) scores between two groups.

Baseline				After Treatment		
Variable	Experimental	Control	*p*	Experimental	Control	*p*
MMSE	28.0 (25–30)	27.0 (26–28)	0.415	28.0 (26.5–30)	27.0 (26–29)	0.181
BI	90.0 (75–100)	95.0 (85–97.5)	0.933	100.0 (90–100)	95.0 (95–100)	0.527
EBI	90.0 (75–90)	90.0 (85–90)	0.438	90.0 (85–90)	90.0 (85–90)	0.603
BBS	48.0 (39–51)	48.0 (47.5–51)	0.188	52.0 (47–54)	52.0 (50.5–54)	0.584
WHODAS 2	31.5 (18–62.6)	27.4 (17.15–36.8)	0.261	22.6 (11.3–50.7)	21.6 (15.3–30.6)	0.740

Explanatory notes: MMSE—Mini-Mental State Examination, BI—Barthel Index, EBI—Extended Barthel Index, BBS—Berg Balance Scale, WHODAS 2—WHO Disability Assessment Schedule 2.0, IQR—Interquartile range, *p*–value.

**Table 3 medicina-59-01669-t003:** Comparison of WHODAS 2 median (IQR) scores between two groups.

	Experimental Group	Control Group	*p*
WHODAS 2			
Baseline	31.5 (18–62.6)	27.4 (17.2–36.8)	0.261
After treatment	22.6 (11.3–50.7)	21.6 (15.3–30.6)	0.740
Four weeks after treatment	20.1 (9.2–43.8)	20.6 (15.1–29.2)	0.996
Cognitive function			
Baseline	16.7 (10.4–66.7)	16.7 (8.3–33.3)	0.470
After treatment	12.5 (8.3–45.8)	12.5 (8.3–27.1)	0.857
Four weeks after treatment	12.5 (6.3–45.8)	12.5 (8.3–25.0)	0.989
Mobility			
Baseline	35 (22.5–75.0)	35 (20–42.5)	0.271
After treatment	20 (15–55)	20 (10–35)	0.577
Four weeks after treatment	20 (10–52.5)	20(10–32.5)	0.881
ADL			
Baseline	37.5 (9.4–65.6)	25 (12.5–46.9)	0.380
After treatment	25 (0–50)	25 (9.4–25)	0.523
Four weeks after treatment	18.8 (0–37.5)	18.8 (9.4–25)	0.965
Relationships			
Baseline	25 (10–50)	25 (10–50)	0.617
After treatment	25 (7.5–35)	20 (15–32.5)	0.981
Four weeks after treatment	15 (5–35)	20 (12.5–27.5)	0.672
Life activities			
Baseline	25 (18.8–50)	25 (17.2–28.1)	0.301
After treatment	25 (12.5–45.3)	21.9 (14.1–25)	0.493
Four weeks after treatment	18.8 (12.5–37.5)	21.9 (14.1–25)	0.753
Participation			
Baseline	53.1 (37.5–73.4)	37.5 (28.1–59.4)	0.052
After treatment	34.4 (29.7–64,1)	37.5 (23.4–45.3)	0.425
Four weeks after treatment	31.3 (26.6–57.8)	34.4 (23.4–46.9)	0.859

Explanatory notes: WHODAS 2.0—WHO Disability Assessment Schedule 2.0, ADL—Activities of daily living, IQR—Interquartile range, *p*–value.

## Data Availability

All research data was securely stored in digital format and protected from loss and unauthorized access. Data may be shared with other researchers upon request.

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
