# Peer review of "Effect of Virtual Reality Therapy on Quality of Life and Self-Sufficiency in Post-Stroke Patients"

_medicina, 2023, doi:10.3390/medicina59091669_

Round 1

Reviewer 1 Report

This manuscript's authors focus on virtual therapy's effects on the quality of life and self-sufficiency in post-stroke patients. The study is very interesting, and the manuscript is well-written. This article may be eligible for publication after correcting minor noticed errors.

Shortcomings:

- In lines 25-26, add when the mentioned WHODAS assessment was performed.

- There are some punctuation errors: lines 37, 38, 53, 114, 171 etc.

- Add the description of the (HRQoL) abbreviation – line 36, a. abbreviation in line 91.

- Consider removing the word  ”balance” from line 65.  

- There is a lack of citation in line 77.

- I suggest moving lines 81-82 to the upper part of the text.

- Add real SD instead of “SD” in the lines 239-240.

- Add a dot after “Table 1”, “Table 2”, "Graph 1” etc.

-Tables 1 and 2 are unclear. Add the precise description. What is in brackets?

All graphs should show the units of the Cartesian axes. Define the whiskers on the graph. Are they SD or SEM?

- Extract the Study Limitation part from the discussion.

- Remove the unnecessary parts of the template like “Patent”,  “Supplementary Materials”, etc.

The style of references should be adapted to the requirements of the publishing house – see template.

Minor editing is needed.

Author Response

Dear reviewer, Thank you very much for your positive evaluation of our manuscript, for the time you devoted to evaluating our manuscript and for all your specific and stimulating comments, which we tried to incorporate carefully.

1.

In lines 25-26, add when the mentioned WHODAS assessment was performed.

Accepted, we have completed the text : Assessment with the WHODAS questionnaire was performed before RHB (T0), after completion of the intervention (T1) and at 4 weeks post-intervention follow-up (T2).

  1. There are some punctuation errors: lines 37, 38, 53, 114, 171 etc.

Accepted, we corrected the punctuation in the mentioned lines, we marked the corrected text in yellow.

  1. Add the description of the (HRQoL) abbreviation – line 36, a. abbreviation in line 91.

Accepted, we added description: Health-Related Quality of Life (HRQoL).

  1. Consider removing the word  ”balance” from line 65.  

Accepted word”balance” was removed.

  1. There is a lack of citation in line 77.

Accepted we added citation : Laver et al. (2017) published that VR therapy alone is not more beneficial than conven- 76 tional therapeutic approaches to improve upper limb function.

  1. I suggest moving lines 81-82 to the upper part of the text.

Accepted, we removed “Given the lack of evidence for the effect of VR on HRQoL and self-sufficiency in post–stroke patients, we focus our study on evaluating the effect of VR in combination with conventional therapy. The primary objective of our randomised trial was to evaluate the effect of conventional rehabilitation combined with VR on improving self-sufficiency and HRQoL in patients after ischemic stroke“ into upper part of „Introduction“( in version after revision lines 43-45)

  1. Add real SD instead of “SD” in the lines 239-240.

Accepted, we added real SD 8.9/ 8.8

  1. Add a dot after “Table 1”, “Table 2”, "Graph 1” etc.

Accepted and added.

  1. Tables 1 and 2 are unclear. Add the precise description. What is in brackets?
  2. All graphs should show the units of the Cartesian axes. Define the whiskers on the graph. Are they SD or SEM?
  3. Extract the Study Limitation part from the discussion.

Accepted , Study limitation was extracted.

12.Remove the unnecessary parts of the template like “Patent”,  “Supplementary Materials”, etc.

Accepted, template  “ Patent” was extracted.

  1. The style of references should be adapted to the requirements of the publishing house – see template.

Accepted, we controlled and corrected all references.

Thank you  very much

Ass. Prof. Dalibor Pastucha, Ph.D., MBA

Reviewer 2 Report

Thank you to the authors for the work done. The article is interesting, and the theme is relevant.

However, I have a few questions and comments.

1.     I ask the authors to make a small correction in line 53: the abbreviation ADL should be put in brackets.

2.     How long from the development of ischemic stroke were patients admitted to the hospital and started receiving treatment? It is described that these data were collected, but they are not reported anywhere in the article.

3.     How many patients were treated only with medication before rehabilitation?

4.     How many patients and how long after the onset of stroke were treated with radiologic endovascular treatment? Accordingly, what is the distribution of patients in the main and comparison groups according to timeliness and type of care provided?

5.     Did patients in the observation (main) group receive VR device treatment and conservative therapy on different days or was there layering? At what time of day was VR device therapy performed? Did the authors consider biorhythms?

6.     The Discussion section is incompletely described. The comparison of the results obtained by the authors with 4 third-party research papers is considered insufficient.

7.     More than 50% of the selected literature references on temporal aspects are not suitable. Accordingly, the key points on the obtained results may not be so disclosed and interpreted.

Author Response

Dear reviewer, thank you very much for your positive evaluation of our manuscript, for the time you devoted to evaluating our manuscript and for all your specific and stimulating comments, which we tried to incorporate carefully.

1I ask the authors to make a small correction in line 53: the abbreviation ADL should be put in brackets.

Accepted and corrected

  1. How long from the development of ischemic stroke were patients admitted to the hospital and started receiving treatment? It is described that these data were collected, but they are not reported anywhere in the article.

Accepted, we added the information and marked it in green in the corrected version of the manuscript

  1. How many patients were treated only with medication before rehabilitation?

Accepted, we added  text: All patients were treated pharmacologically prior to the start of rehabilitation, and during their stay in the rehabilitation facility they underwent regular medical exami-nations at least once a week.

  1. How many patients and how long after the onset of stroke were treated with radiologic endovascular treatment? Accordingly, what is the distribution of patients in the main and comparison groups according to timeliness and type of care provided?

Accepted, we added text: All patients were treated according to the European Stroke Organization (ESO) standard operating procedure based on the Grading of Recommendations, Assessment, Devel-opment and Evaluations (GRADE) framework.

  1. Did patients in the observation (main) group receive VR device treatment and conservative therapy on different days or was there layering? At what time of day was VR device therapy performed? Did the authors consider biorhythms?

Accepted, we added text: Patients enrolled in the study received exercise in VR due to organizational reasons of the rehabilitation facility in the afternoon from 3 hours after the completion of all treatments

  1. The Discussion section is incompletely described. The comparison of the results obtained by the authors with 4 third-party research papers is considered insufficient.

Accepted we added into “Discussion” text: The level of education, the level of comorbidity, as well as socioeconomic situation, occupational status and potential return to work are also related to quality of life. In our study, most probands were living in a relationship with a partner and all patients planned to return to their home environment. Hayes et al. (2003) reported in their study that up to 25 % of patients after stroke remain in institutional care, and in their analyses, women were more dependent on ADL, less likely to go unassisted, and lived in nursing homes [7]. Hu et al. (2017) concluded that domain values of cognition level, mobility, and WHODAS 2.0 sum score were good predictors of institutionalization [51]. The au-thors of the studies also point out the influence of local social welfare policy, disparities in social support and disability service system [51]. Lee et al. (2022) in their study re-ported that perceived social support is positively correlated with motivation to recover [19]. The higher the perceived social support, the stronger the stroke patient's motiva-tion to recover and the higher the likelihood of lower disability rates [19].

  1. More than 50% of the selected literature references on temporal aspects are not suitable. Accordingly, the key points on the obtained results may not be so disclosed and interpreted.

Accepted , we removed from original paper cittations  of No 9, 39, 53 ( as citated in original version).

Thank you  very much

Ass. Prof. Dalibor Pastucha, Ph.D., MBA
